# miR-205 in Breast Cancer: State of the Art

**DOI:** 10.3390/ijms22010027

**Published:** 2020-12-22

**Authors:** Ilaria Plantamura, Alessandra Cataldo, Giulia Cosentino, Marilena V. Iorio

**Affiliations:** Molecular Targeting Unit, Research Department, Fondazione IRCCS Istituto Nazionale dei Tumori, 20133 Milan, Italy; ilaria.plantamura@istitutotumori.mi.it (I.P.); alessandra.cataldo@istitutotumori.mi.it (A.C.); giulia.cosentino@istitutotumori.mi.it (G.C.)

**Keywords:** miR-205, breast cancer, oncogenic pathways, therapy

## Abstract

Despite its controversial roles in different cancer types, miR-205 has been mainly described as an oncosuppressive microRNA (miRNA), with some contrasting results, in breast cancer. The role of miR-205 in the occurrence or progression of breast cancer has been extensively studied since the first evidence of its aberrant expression in tumor tissues versus normal counterparts. To date, it is known that the expression of miR-205 in the different subtypes of breast cancer is decreasing from the less aggressive subtype, estrogen receptor/progesterone receptor positive breast cancer, to the more aggressive, triple negative breast cancer, influencing metastasis capability, response to therapy and patient survival. In this review, we summarize the most important discoveries that have highlighted the functional role of this miRNA in breast cancer initiation and progression, in stemness maintenance, in the tumor microenvironment, its potential role as a biomarker and its relevance in normal breast physiology—the still open questions. Finally, emerging evidence reveals the role of some lncRNAs in breast cancer progression as sponges of miR-205. Here, we also reviewed the studies in this field.

## 1. Introduction

MicroRNAs (miRNAs) are small non-coding single-stranded RNAs (19–25 nucleotides), and their main function is to downregulate their target gene binding to the 3′UTR of mRNAs. In cancer, miRNAs are classified as oncogenes or oncosuppressors based on their capability to target tumor suppressor genes or oncogenes, respectively. It is well known that miRNAs, downregulating their targets, are involved in the biological steps of tumor progression such as cancer initiation, proliferation, metastasis, epithelial-to-mesenchymal transition (EMT), stemness maintenance, and therapeutic response and resistance. Therefore, dysregulated miRNAs could be used as biomarkers for diagnosis and potential targets for cancer treatment [1].

In this review, we focused our attention on the role of miR-205 in human breast cancer disease. MiR-205 is located on chromosome 1q32.2 and its structure is highly conserved, and it has been mainly described as an oncosuppressive miRNA in breast cancer [1].

Breast cancer is the most widespread female malignancy in the world and it is divided into three different subtypes by immunohistochemistry classification: estrogen/progesterone positive (ER+/PR+), human epidermal growth factor receptor 2 positive (HER2+) and triple negative breast cancer (TNBC); and in five different subtypes by molecular classification: Luminal A, Luminal B, HER2-enriched, basal-like breast cancer and normal-like [2]. In 2005, we reported for the first time the aberrant expression of miR-205 in breast cancer tissues, which is also negatively associated with the presence of vascular invasion [3]. A couple of years later, Sempere L.F. and colleagues provided a deeper evaluation of the expression pattern of this microRNA in the breast tissues by in situ hybridization (ISH), revealing that it is normally confined to a specific cell subpopulation, the myoepitelial/basal cell layer; this particular accumulation is lost in the neoplastic counterpart [4]. Thus, many studies reveal that miR-205 has a decreasing expression from the less aggressive subtype, the ER+/PR+ breast cancer, to the more aggressive subtype, the TNBC. Indeed, miR-205 is overexpressed in ER+/PR+ breast cancer in comparison with HER2+ breast cancer, and in TNBC it is downregulated compared to the other subtypes [1]. MiR-205 was not only downregulated in tumors versus normal tissue, but a lower expression was also reported in metastases in comparison with normal tissues, and in metastatic lymph nodes in comparison with primary tumors [5,6]. Moreover, it was demonstrated that miR-205 expression was decreased in inflammatory breast cancer compared with non-inflammatory breast cancer [7]. All these findings suggest that the loss of this miRNA promotes cancer progression. Moreover, Berber U. and colleagues demonstrated that lower miR-205 levels potentially predict lymph node metastasis in TNBC patients [8]. Consistently, miR-205 downregulation in tumor tissues has been associated with poor prognosis in early breast cancer, including within specific breast tumor subtypes, such as TNBC and HER2+ malignances [9,10,11,12]. Furthermore, it is known that miRNAs could be found in the circulation of cancer patients. Concerning miR-205, it is downregulated in the serum of breast cancer patients compared to healthy people, suggesting that miR-205 might be useful for the clinical diagnosis of breast cancers [13,14]. However, contrasting results were reported by different studies; circulating levels of miR-205 were found significantly upregulated in the serum of breast cancer patients and associated, along with miR-19a, with resistance to neoadjuvant chemotherapy in Luminal A breast cancer [15,16]. In contrast, a very recent report reveals instead that serum levels of miR-205 and -375 are associated with responsiveness to neoadjuvant chemotherapy in Luminal A tumors, whereas miR-205 and -21 serum levels are associated with responsiveness to neoadjuvant chemotherapy in Luminal B tumors [17]. This discrepancy might be related to different methods of normalization, which still represents an issue for circulating microRNAs and underlines how these findings need to be further investigated to support a potential use of circulating miR-205 as a biomarker in breast cancer patients. In parallel to expression profiles, functional studies have been performed to assess the biological role of this molecule and to dissect the regulatory pathways it is able to regulate. Here, we summarize miR-205’s impact on the major oncogenic pathways in breast cancer. Main miR-205 direct targets are reported in Table 1.

## 2. Role of miR-205 in Cell Growth, Proliferation and Responsiveness to Therapy

In 2012, our group investigated the miR-205 spectrum of action in TNBC. Firstly, we proved that this miRNA exerts an oncosuppressive function by directly targeting E2F Transcription Factor 1 (E2F1) and Laminin Subunit Gamma 1 (LAMC1) [18]. E2F1 is a well-known master regulator of cell cycles, whereas LAMC1 belongs to the laminin family and has been associated with cell adhesion, proliferation and migration [34]. Interestingly, we also demonstrated that the known oncosuppressor p53 is an important transcription factor for the miRNA [18]. MiR-205 is, in fact, directly transactivated by p53 and this axis could prevent breast cancer cell cycle progression. It is well known that p53 is indeed able to induce both cell cycle arrest and cell death and it participates in various steps of differentiation and development [35].

More recently, it has been shown that miR-205 directly targets Krüppel-like factor 12 (KLF12) to reduce the progression of breast cancer (BC) basal-like malignances [19]. In particular, the ectopic expression of miR-205 in basal B TNBC MDA-MB-468 cells leads to a significant decrease in cell invasion and apoptosis. The miR-205-KLF12 regulation could be crucial for BC basal-like disease. KLF12 is indeed able to regulate different biological processes of cancer cells, such as cell growth, cell apoptosis, differentiation, proliferation and angiogenesis [36,37].

MiR-205 has been also demonstrated to negatively regulate the oncogene Nuclear factor I/B (NFIB) in ER+ breast cancer [20]. Chen H et al. demonstrated that Nuclear factor I/B (NFIB) was upregulated in ER+ breast cancer versus normal tissue and that NFIB promoted MCF-7 cell cycle progression and proliferation in in vitro experiments. NFIB is overexpressed in different tumor subtypes and it affects tumor progression and metastasis [38].

MiR-205 also participates in the cell proliferation of ER+/PR+ breast cancer; in fact, studies have demonstrated that angiomotin (AMOT), an adaptor protein regulating tight junctions, activates the ERK1/2 pathway to drive cell proliferation in ER+ breast cancer, and that miR-205 interferes with this mechanism by directly targeting AMOT in MCF7 cells [21,39]. However, it is not infrequent to find contrasting evidence regarding the role of a miRNA in the same pathology; miR-205 is no exception. In fact, Qiu C. and colleagues reported that high levels of this miRNA are associated with an increased proliferation and invasive ability in MCF7 cells and with a reduced survival time in breast cancer patients [40].

MiR-205 modulation also impacts breast cancer’s responsiveness to therapy. For example, miR-205 was seen to reduce gemcitabine resistance by targeting endoplasmic reticulum protein 29 (ERp29) levels in breast cancer [22]. In contrast, Cai Y. et al. have demonstrated that miR-205 has a synergistic effect with docetaxel in breast cancer both in in vitro and in vivo models. Indeed, the re-introduction of miR-205 in MDA-MB-231 and MCF7 combined with docetaxel inhibited colony formation ability and increased sensitivity to the treatment in both cell lines. Moreover, MDA-MB-231/miR-205 stable clones xenografted in vivo showed a reduction in tumor growth alone and a higher decrease in combination with docetaxel [41]. In 2009, we demonstrated that miR-205 directly targets HER3 in HER2+ breast cancer cell lines, leading to impairment of the Akt-mediated survival pathway, and improved responsiveness to TKI inhibitors such as Gefitinib and Lapatinib [23]. Furthermore, we have recently demonstrated that miR-205 also enhances sensitivity to trastuzumab by HER3 targeting and the impairment of Akt signaling in HER2+ breast cancer models [11]. The direct targeting of HER3, HER2 favorite cognate receptor, was also confirmed by other groups, either in breast cancer models or other tumor types, such as prostate cancer [32,42]. For instance, miR-205-mediated HER3 targeting also leads to the impairment of migration and invasion and the induction of apoptosis in the luminal MCF7 cell line [24]. MiR-205 was also seen to act on the downstream effectors of the HER2 signaling pathway. In fact, Takeno T. and colleagues recently demonstrated that miR-205 directly binds the 3′ UTR of chloride voltage-gated channel 3 (CLCN3) mRNA. CLCN3 is induced by HER2 upregulation and is thought to regulate cell proliferation and apoptosis through the modulation of cell volume. Consequently, by using HER2-overexpressing breast epithelial cell line MCF10A-ErbB2, they showed that miR-205 re-introduction or CLCN3 silencing reduced 3D spheroid proliferation [26]. It is interesting to note that the same group demonstrated that HER2 epigenetically represses miR-205 transcription via the Ras/Raf/MEK/ERK pathway, likely to be a strategy exploited by tumor cells to counteract miR-205 oncosuppressive activity [43,44]. Consistently, we observed a strong downregulation of miR-205 in HER2+ breast cancer cell lines and in the tumor originating from MMTV-neu transgenic mice (unpublished data).

Moreover, miR-205, together with miR-125b, inhibited HER3 expression and the activation of downstream signaling, decreasing cell proliferation and enhancing the G1 phase of cell cycle in a HER2 positive breast cancer model. The expression of miR-205 and miR-125b also promotes the sensitivity to trastuzumab and to paclitaxel in HER2 positive breast cancer cell lines [45]. Downregulation of HER2/HER3 signaling by miR-205 was also demonstrated by Wang S. et al. Indeed, they reported that an HDAC inhibitor, entinostat, specifically enhances the expression of miR-125a, miR-125b, and miR-205, which act together to downmodulate HER2/HER3, thus inducing apoptosis in breast cancer cells [25].

In addition, a downregulated expression of miR-205 was found in a patient-derived xenograft (PDX) model from HER2 positive tumors resistant to trastuzumab [46].

## 3. Role of miR-205 in Epithelial-to-Mesenchymal Transition

During epithelial-to-mesenchymal transition, cancer cells acquire a mesenchymal–like phenotype that enhances migration and invasion capability, with the consequence of an improved metastatic potential [47].

In 2008, Gregory and colleagues conducted a pivotal study describing the capability of the miR-205 and miR-200 family to inhibit EMT by directly targeting zinc finger E-box binding homeobox 1 (ZEB1) and Smad interacting protein 1 (SIP1) [27]. This report has really been seminal for a huge number of following studies performed in the last 10 years, considering the crucial role of the EMT process in the tumor’s acquisition of features of aggressiveness, invasiveness, and resistance to therapies. In fact, in 2014 Zhang P. and colleagues demonstrated that miR-205 is downmodulated in radioresistant subpopulations of breast cancer cells and that it enhances radiosensitivity. The mechanism related to this phenomenon is the downregulation of DNA damage repair triggered by miR-205 re-introduction, which targets ZEB1 and the ubiquitin-conjugating enzyme Ubc13. Moreover, in this work, the authors therapeutically delivered miR-205 mimics via nanoliposomes, resulting in the higher sensitivity to radiation in a breast cancer xenograft model [12]. The miR-205/ZEB1 axis in the MDA-MB-231 TNBC cell line has later been confirmed by Lee J.Y. and colleagues [48]. The authors also demonstrated that miR-205 is epigenetically modulated by the polycomb protein Mel-18, which impairs the DNMT-mediated promoter methylation, leading to an increase in miR-205 expression and the silencing of direct targets ZEB1/2.

Consistent with an epigenetic regulation of miR-205 expression, treatment with 5-Aza-2′-Deoxycytidine (5-AZA) and Trichostatin A (TSA) de-represses miR-205 expression in MDA-MB-231 TNBC cells [28]. Additionally, the authors showed the direct regulation of an additional target, high mobility group box 3 (HMGB3), also linked to EMT.

Recently, Wang’s group demonstrated that miR-205 targets high mobility group box 1 (HMGB1) in the MDA-MB-231 TNBC cell line and this axis suppresses the EMT process; in particular, they showed that the re-introduction of miR-205 inhibited HMGB1/RAGE (receptor for advanced glycation end products) expression, resulting in the suppression of tumor invasion and EMT [29].

Moreover, miR-205 is inhibited by the transglutaminase 2 (TG2) in order to induce EMT in breast cancer cell lines; the GTP binding activity of TG2 represses miR-205 expression, thus allowing ZEB1 expression and function in in vitro assays [49]. In addition, in in vivo experiments, stable miR-205 overexpression reduced bone metastases generated by intracardiac injection of MCF7 cells overexpressing an activated TG2. All these studies confirm the central role of miR-205 in the regulation of EMT.

## 4. Role of miR-205 in Stemness Maintenance

Breast cancer stem cells (BCSCs) are cells with a high tumor formation capability that present the self-renewal characteristics of stem cells [50]. These cells are CD44+/CD24- and have a key role in the different steps of tumor initiation, progression and therapy response. Increasing evidence confirms that miRNAs can modify the stemness of BCSCs, altering breast cancer progression in terms of tumor formation, self‑renewal, differentiation, metastasis, tumorigenicity and chemotherapy resistance [51]. This evidence suggests that a miRNA-based therapy might also be efficient in impacting tumor occurrence and progression by affecting cancer stem cells, known to strongly contribute to tumor recurrence and resistance to therapies [52].

In the context of miR-205, recent studies have demonstrated that this miRNA is downregulated in TNBCs, which, together with HER2+ malignances, are the tumors with the highest number of BCSCs [53]. It has also been reported that miR-205 levels are negatively correlated with stemness in breast cancer [30]. In particular, miR-205 overexpression reduces the expression of CD44+/CD24- markers in MDA-MB-231 TNBC cells by inhibiting the expression of its target gene RUNX Family Transcription Factor 2 (RunX2), both in vitro and in vivo. Indeed, in different BC cell lines the expression of miR-205 and RUNX2 are inversely correlated. The overexpression of RUNX2 and its role in tumor progression in breast cancer have been demonstrated in different studies [54]. Moreover, RUNX2 overexpression in the MCF7 breast cancer cell line induced EMT pathways [55]. Thus, the miR-205/RUNX2 axis leads to the inhibition of the EMT process, invasion, migration and stemness maintenance in breast cancer.

Moreover, Xiao Y et al. demonstrated that, in contrast to integrin α5 (ITGA5) level, the expression of miR-205 is lower in basal mesenchymal-like TNBC cells, and that miR-205 suppresses TNBC tumor growth, migration and cancer stem cells by downregulating ITGA5 levels [31]. ITGA5 is a member of the family of integrins, which play a role in cell adhesion, proliferation, migration, invasion and cancer metastasis [56].

Surprisingly, De Cola and colleagues have reported the upregulation of miR-205 in patient-derived breast cancer stem cells, with the consequent downregulation of Epidermal Growth Factor Receptor (EGFR) and HER2 and resistance to Lapatinib [57]. In 2018, the same group demonstrated that miR-205 silencing reduces BCSC metastatic potential in vitro and in vivo by modulating the EMT pathway [58]. Even though this opposite mechanism might be limited to BCSCs, it is not consistent with previously reported data, and it needs to be further explored for potential consequences on responsiveness to anti-HER2 therapies.

## 5. Role of miR-205 in Shaping the Tumor Microenvironment

MiRNAs have been shown to act not only directly on tumor cells, but also on components of the surrounding microenvironment, where they play crucial roles in molding a pro or anti-tumorigenic milieu. For instance, miR-205 is also implicated in the process of angiogenesis, crucial not only for tumor survival but also for cancer cell dissemination at secondary sites. Wu H. and colleagues demonstrated that MDA-MB-231 cell line transfected with miR-205 shows reduced in vivo lung metastatic potential, at least in part through vascular endothelial growth factor A (VEGF-A) repression [32]. VEGF is not only implicated in the angiogenesis process, but it also regulates different pathways. For example, in breast cancer patients, low expression levels of VEGF-A and fibroblast growth factor 2 (FGF2) lead to a better response to neoadjuvant chemotherapy. In this context, Hu Y.’s data suggest that miR-205 is able to target VEGF-A and FGF2 in breast cancer cells, impairing the PI3K/AKT signaling and thus triggering apoptosis and the restoration of chemo-sensitivity in resistant BC cells [33].

Moreover, in the specific context of tumor stromal cells, miR-205 is found downregulated in breast cancer-associated fibroblasts (CAFs) compared with normal fibroblasts (NFs) [59]. This is further supported by the work of Du YE et al., which shows that reduced levels of miR-205 in normal fibroblasts induced CAF phenotype and YAP1-mediated angiogenesis [60].

Finally, it was demonstrated that miR-205, together with miR-31, is able to induce a repression of metastatic potential when MDA-MB-231 TNBC cells are treated with extracellular microvescicles (MVs) from mesenchymal stromal cells (MSCs) containing these two miRNAs [61]. This effect is mediated by the downregulation of a common target, leading to the suppression of migration, invasion and proliferation of breast cancer cells.

The role of miR-205 in the tumor microenvironment is still not completely elucidated in breast cancer; more studies in this field are needed to better understand whether this miRNA might counteract the known escape mechanisms triggered by tumor-associated stromal cells.

## 6. Role of miR-205 in Normal Breast Physiology

Despite the well reported oncosuppressive function of miR-205 in established breast tumors, the evidence concerning the role of this miRNA in normal breast physiology is a bit controversial. MiR-205 expression is modulated during mammary gland morphogenesis; indeed, it increases in both basal and luminal epithelium during pregnancy and lactation with a subsequent increase in expression during late involution [62].

Surprisingly, whereas previous evidence associated miR-205 with stem cell progenitors of normal breast tissue [63], the impaired expression of this microRNA in Wild Type (WT) mouse models did not cause any defect in the mammary gland morphogenesis; instead, it promoted ductal outgrowth, the spontaneous formation of hyperplatic lesions and, in 60% of miR-205 KD mice, focal mammary carcinoma [64].

Furthermore, whereas in Balb/c mice the genetic knockout of miR-205 caused neonatal lethality with severely compromised epidermal and hair follicle growth, in a different strain (FVB), Rosen and collaborators did not observe a lethal phenotype [65]. However, a more recent paper reported that genetic loss of miR-205 did cause a reduced mammary regenerative potential, but only under stress conditions [66]. In particular, a significant difference in the gland outgrowth was observed only in mammary epithelial cell (MEC) transplantation experiments, with a reduced number (1000) of cells in miR-205 WT versus KO mice. Vice versa, no effect was observed by transplanting 100,000 cells into the mammary fat pad of the mice differing for miR-205 status.

Consistent with the report from Chao CH et al. [64] our own data indicate instead that genetic loss of miR-205 induces mammary gland hyperplasia (unpublished data).

The role of miR-205 in normal physiology is currently an open question that needs further investigation.

## 7. MiR-205 and Long Non-Coding RNAs

Long noncoding RNAs (lncRNAs) belong to a large class of non-coding RNAs, >200 nucleotides long, that are involved in different biological processes, including cancer cell invasion and metastasis [67]. Recently, many studies have shown the association of lncRNAs with both breast cancer biology and miRNA’s expression, function and regulation [68]. Indeed, lncRNAs can selectively bind miRNAs acting as sponges or decoys, thus impeding their inhibitory activity on target mRNAs. Together, lncRNAs and miRNAs could be used as predictive biomarkers or as therapeutic tools [69].

Recent studies have reported that some lncRNAs could act as sponges on miR-205, modifying several mechanisms in breast cancer cells, such as EMT. For example, lncRNA-ROR has been described to act as an endogenous RNA competing with miR-205, thus preventing the degradation of miRNA target molecules and favoring the EMT process [70]. Moreover, in 2017, Grelet S. et al. demonstrated that lncRNA-PNUTS interacts with and regulates miR-205. In more detail, lncRNA-PNUTS transiently sponges miR-205 to induce an upregulation of ZEB factors with a consequent regulation of EMT and tumor progression. Indeed, the transfection of wild-type lncRNA-PNUTS induces EMT phenomenon, downregulating E-cadherin and upregulating ZEB1; conversely, the co-transfection with miR-205 rescues this effect [71].

LncRNAs can also affect response to therapy. Indeed, Zhang H. Y. and colleagues showed that lncRNA-ROR induced tamoxifen-resistance by miR-205-5p suppression [72]. Moreover, another lncRNA, named CCAT2, was shown to target miR-205 in TNBC, inducing an oncogenic phenotype [73]. Surprisingly, another study revealed that miR-205-5p regulation by lncRNA FGF14-AS2 inhibited proliferation, migration and invasion and promoted apoptosis in breast cancer. In addition, the authors demonstrated an upregulation of miR-205 in high stage breast cancer [74], suggesting an oncogenic role of this miRNA in breast cancer progression.

The field of lncRNAs is still an unexplored world; consequently, more studies could be necessary to completely elucidate their role in the modulation of miR-205 expression, and certainly of other miRNAs.

## 8. Concluding Remarks

MicroRNAs are small RNA molecules aberrantly expressed in specific diseases, including human cancer. MiRNA signatures associated with tumor diagnosis and predictions of outcome and/or responsiveness to therapy have been identified. In particular, miR-205, detectable in both neoplastic cells and serum of breast cancer patients, has been associated with better outcomes in different subtypes of breast cancer and following different therapeutic treatments, indicating that the expression levels of this miRNA might serve as a prognostic and/or predictive biomarker.

In addition, the functional role exerted by miRNAs in tumor occurrence and progression has raised the fascinating possibility of an innovative miRNA-based therapeutic approach. Based on the discoveries regarding the role of miR-205 in breast cancer cell intrinsic and extrinsic mechanisms, it is clear that this miRNA could be a valid candidate as a therapeutic tool for breast cancer patients in the future. The feasibility of miRNAs in therapy, alone or in combination with other drugs, has not been demonstrated yet. Further studies are needed in this field to minimize the toxicity and the off-target effects. Moreover, it is necessary to find a specific system to deliver the microRNAs directly to the tumor. However, despite the contrasting ideas on miRNA-based therapy, the majority of the studies demonstrate that miR-205 plays an oncosuppressive role in breast cancer, acting both on neoplastic cells and tumor microenvironment, and its re-introduction in tumors might be exploited to counteract tumor growth and to improve responsiveness to breast cancer therapies, mainly anti-HER2 treatments. Accordingly, Lai et al. have created a mathematical model describing a system biology-based investigation of combining the feasibility of various miRNAs as monotherapy or adjuvant therapy for various cancers [75]. Among these microRNAs, the combination of miR-205 and miR-342 seems to have a high impact on chemotherapy response.

Figure 1 summarizes miR-205 activity on the main oncogenic pathways altered in breast cancer disease.

## Figures and Tables

**Figure 1 ijms-22-00027-f001:**
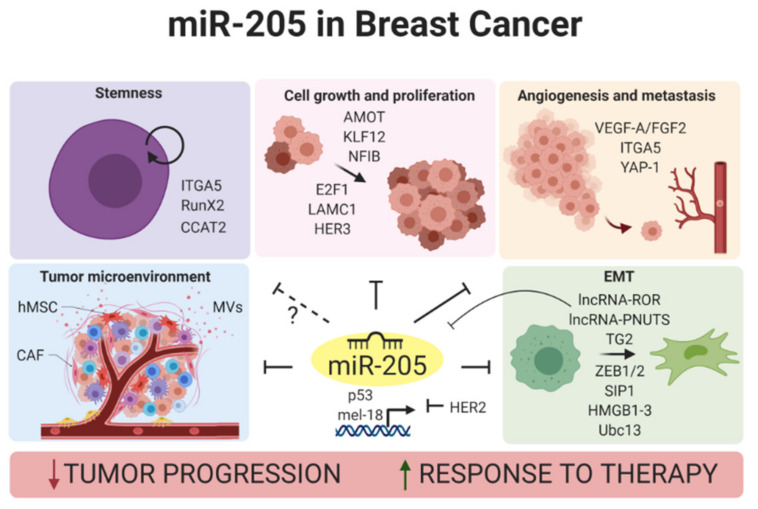
Overview of miR-205 oncosuppressive activity in breast cancer (created with BioRender.com).

**Table 1 ijms-22-00027-t001:** Summary of miR-205 main targets and the consequent oncosuppressive effect in breast cancer cells.

miR-205 Target	Target Function	Effects of miR-205 Targeting in Breast Cancer
E2F1	transcription factor	Reduction in cell proliferation, cell cycle progression and clonogenic potential in vitro, and inhibition of tumor growth in vivo [18]
LAMC1	matrix glycoprotein	
KLF12	transcription factor	Reduction in proliferation, invasion and increase in apoptosis [19]
NFIB	transcription factor	Reduction in cell cycle progression, cell proliferation and increase in apoptosis [20]
AMOT	angiostatin binding protein	Inhibition of cell growth [21]
ERp29	chaperone protein	Increase in gemcitabine sensitivity [22]
HER3	tyrosine-kinase receptor	Increase in sensitivity to TKI inhibitors and Trastuzumab in HER2+ malignances [11,23,25]
CLCN3	chloride voltage-gated channel	Reduction in 3D spheroid proliferation [26]
ZEB1	zinc finger protein	Inhibition of EMT [27]
SIP1	zinc finger protein	
Ubc13	ubiquitin E2 conjugating enzyme	Enhancement of cell radiosensitivity [12]
HMGB3	chromatin-binding protein	Inhibition of EMT [28]
HMGB1	chromatin-binding protein	Inhibition of EMT and reduction in tumor invasion [29]
RunX2	transcription factor	Inhibition of EMT, invasion, migration and stemness maintenance [30]
ITGA5	integrin protein	Suppression of TNBC tumor growth, migration and cancer stem cells [31]
VEGF-A	growth factor	Reduction in in vivo lung metastasis, induction of apoptosis and better response to neoadjuvant chemotherapy [32,33]
FGF2	growth factor	Induction of apoptosis and better response to neoadjuvant chemotherapy [33]

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
