# Peer review of "miR-205 in Breast Cancer: State of the Art"

_ijms, 2020, doi:10.3390/ijms22010027_

Round 1

Reviewer 1 Report

The manuscript summarizes the role of miR-205 in cell growth, responsiveness to therapy, EMT, stemness maintenance, shaping the tumor microenvironment in breast cancer. Its role in normal breast physiology was also discussed. On top of that, the authors also discuss how the sponge of miR-205 affects the pathogenesis of breast cancer.

There are several areas which require clearer explanations:

  1. In lines 58-61, the authors should try to explain why contradictory results were observed. Is it related to different types of samples used? or due to samples from different races or different detection methods were used?
  2. In lines 61, it is not understandable how the contradictory results from different studies support the statement: miR-205 could be used as a biomarker in breast cancer patients. A clinically used biomarker must have little variation across human subjects.
  3. Line 70 to 74, authors are talking about the interaction between miR-205 and p53, KLF12 and NFIB. However, authors have not related these pathways to cell growth or proliferation, which is the main focus of this paragraph.
  4. In line 133, authors suggested BSCSs may be potential targets for a miR-based therapy. What kind of miR-based therapy? and how to target ? it is not explained and no reference is quoted for such a statement. Futhur explanation is required.
  5. In line 139, thus, the axis leads to the inhibition of EMT...... The causal relationship between the RUNX2 and EMT, invasion, migration are not explained.
  6. in line 182, regarding "the formation of mammary lesions", the characteristics or significant of that could be elaborated more.
  7. in line 187,"only" in the transplantation experiments, how is it related to the stress conditions mentioned? and what are the other experiments?
  8. in line 23 " This is a controversial result..... " When lncRNA FGF14 binds to miR205, the amount of effective miR205 decreases, resulting in reduced proliferation, migration and invasion. High stage breast cancer is aggressive, with high proliferation rate and high invasiveness. An upregulation of miR205 in high grade breast cancer seems logical. Why is it controversial?
  9. Searching pubmed with ' miR-205 + breast cancer', there are a total of 102 results. At least 10 more papers related to miR-205 and breast cancer could be included in the review. 

Minor comments:

Line 117: Impacts, better indicates whether it suppresses or triggers 

Line 120: resulted in a tumor invasion and EMT suppression. Does it mean resulted in a (tumor invasion) and (EMT suppression) or resulted in a suppression of tumor invasion and EMT?

Line 171: Field vs filed

Author Response

We thank the reviewer for her/his useful suggestions and comments. We have tried to improve our manuscript by following the indications. Please find a point-by-point reply here below:

REVIEWER 1

The manuscript summarizes the role of miR-205 in cell growth, responsiveness to therapy, EMT, stemness maintenance, shaping the tumor microenvironment in breast cancer. Its role in normal breast physiology was also discussed. On top of that, the authors also discuss how the sponge of miR-205 affects the pathogenesis of breast cancer.

There are several areas which require clearer explanations:

  1. In lines 58-61, the authors should try to explain why contradictory results were observed. Is it related to different types of samples used? or due to samples from different races or different detection methods were used?

We thank the reviewer for raising this issue.

The different miR-205 expression in serum of breast cancer patients in the reported studies is more likely related to a different normalization method (miR-16 or U6 or snord) more than suptypes (which are represented in the different cohorts) or race, even though we cannot exclude a differential expression. Some of the molecules used to normalize miR-205 expression were not probably a reliable housekeeping: indeed finding a valid housekeeping molecule to normalize circulating microRNA expression is still quite an issue. We added a sentence in the text (lanes 69-72).

  1. In lines 61, it is not understandable how the contradictory results from different studies support the statement: miR-205 could be used as a biomarker in breast cancer patients. A clinically used biomarker must have little variation across human subjects

The reviewer is right, our sentence was confusing since referred to tissue miR-205 expression instead of ct-miR-205, thus resulting out of context. We modified the text (lanes 71-72).

  1. Line 70 to 74, authors are talking about the interaction between miR-205 and p53, KLF12 and NFIB. However, authors have not related these pathways to cell growth or proliferation, which is the main focus of this paragraph.

             We thank the reviewer for this comment. We better described the pathways cited in the text as

             affected by miR-205, and specified the functional role in the context of cell growth and proliferation

             (lanes 82-96).

  1. In line 133, authors suggested BSCSs may be potential targets for a miR-based therapy. What kind of miR-based therapy? and how to target ? it is not explained and no reference is quoted for such a statement. Futhur explanation is required. We thank the reviewer for this comment. The paragraph wasn’t indeed very clear. We meant that, given the capability of miRNAs to regulate stemness properties, and given the role of cancer stem cells in tumor occurrence, progression and responsiveness to therapies, a miRNA-based therapy where we modulate miRNA expression might have a significant impact on the disease outcome. We modified the sentence in the text (lanes 182-185) and added a reference (49).
  2. In line 139, thus, the axis leads to the inhibition of EMT...... The causal relationship between the RUNX2 and EMT, invasion, migration are not explained. We thank the reviewer for this comment. We better explained in the text (lanes 191-195) the correlation between RUNX2 and breast cancer features of aggressiveness.
  • in line 182, regarding "the formation of mammary lesions", the characteristics or significant of that could be elaborated more. We better explained in the text (lanes 245-247).
  • in line 187,"only" in the transplantation experiments, how is it related to the stress conditions mentioned? and what are the other experiments? We modified the text providing more details (lanes 252-256).
  • in line 23 " This is a controversial result..... " When lncRNA FGF14 binds to miR205, the amount of effective miR205 decreases, resulting in reduced proliferation, migration and invasion. High stage breast cancer is aggressive, with high proliferation rate and high invasiveness. An upregulation of miR205 in high grade breast cancer seems logical. Why is it controversial? Thanks for this comment. The result is controversial since miR-205 is reported to be an oncosuppressive miRNA in most cases. We modified the text (lanes 280-284).
  1. Searching pubmed with ' miR-205 + breast cancer', there are a total of 102 results. At least 10 more papers related to miR-205 and breast cancer could be included in the review. According to reviewer’s request, we increased the references concerning miR-205 in breast cancer.

Minor comments:

Line 117: Impacts, better indicates whether it suppresses or triggers. We specified in the text (lane 166).

Line 120: resulted in a tumor invasion and EMT suppression. Does it mean resulted in a (tumor invasion) and (EMT suppression) or resulted in a suppression of tumor invasion and EMT? We specified in the text (lanes 167-169).

Line 171: Field vs filed. Thank you, we corrected the typing error.

Reviewer 2 Report

This review, from Plantamura and colleagues, gives an overview of the role of miR-205 in breast cancer. I have found this review easy to read, interesting and very well written.

Overall, I have really appreciated the work of the authors summarizing all the bibliography regarding solid/consistent data of the selected miRNA in breast cancer.

Minor comments:

  1. Progesterone receptor is commonly known as PR, instead of PgR.

Author Response

We thank the reviewer for her/his useful suggestions and comments. We have tried to improve our manuscript by following the indications. Please find a point-by-point reply here below:

REVIEWER 2.

This review, from Plantamura and colleagues, gives an overview of the role of miR-205 in breast cancer. I have found this review easy to read, interesting and very well written.

Overall, I have really appreciated the work of the authors summarizing all the bibliography regarding solid/consistent data of the selected miRNA in breast cancer.

Minor comments: Progesterone receptor is commonly known as PR, instead of PgR.

We really thank the reviewer for the appreciation of our work.

We corrected Progesteron Receptor abbreviation in PR.

Reviewer 3 Report

This paper discussed role of miRNA-205 in breast cancer. However, this paper suffers from low number of clinical proofs concerning its utility for breast cancer patients. Some tables and additional figures, which could improve paper quality are missed. 

Author Response

We thank the reviewer for her/his useful suggestions and comments. We have tried to improve our manuscript by following the indications. Please find a point-by-point reply here below:

REVIEWER 3:

This paper discussed role of miRNA-205 in breast cancer. However, this paper suffers from low number of clinical proofs concerning its utility for breast cancer patients. Some tables and additional figures, which could improve paper quality are missed.

Response:

We thank the reviewer for the comment regarding the clinical role and the utility of miR-205 in breast cancer patients.

In general, the utility of miRNAs in therapy has not been demonstrated yet, both alone and in combination with standard therapies. To render miRNA-based therapy feasible, it is necessary to minimize the toxicity and the off-target effects. Moreover, it is necessary to find a specific system to deliver the microRNA directly to the tumor. We specified the sentence in the conclusion (lanes 302-305).

A strategy might be the conjugation of miRNA-carrying nanoparticles with specific antibodies, avoiding that microRNAs affect healthy cells and modulate other target mRNAs. Several trials testing miRNA-based therapies are on-going (www.clinicaltrials.gov), even in different tumor types, but none of them is testing miR-205. Despite these issues, miR-205 restoration in breast cancer does have a high potential as a therapeutic tool, as clearly suggested by its anti-cancer activity exerted modulating cell proliferation, EMT process, cell cycle, apoptosis and tumor growth. In this context, we agree with the reviewer that this review lacks some data regarding the role of miR-205 reintroduction in response to therapy, which we have now introduced in the text (lanes 107-112; 131-140; 309-312). However, regarding a clear proof of a potential miR-205-based therapy, unfortunately to date there are few studies about miR-205 restoration in in vivo breast cancer models, mainly by creating stable clones expressing miR-205. Only few studies have been performed using nanotechnologies to reintroduce the miRNA in the tumor, but these studies have been performed in prostate models (1-4). Since our review is focused on breast cancer, we can only speculate a hypothetical role of this microRNA in clinic practice in breast cancer, based on the results obtained from preclinical studies.

Finally, to improve the quality of this review we add a table that summarizes miR-205 main targets and the consequent oncosuppressive effect in breast cancer cells as requested by the reviewer.

  • Nagesh, P.K.B.; Chowdhury, P.; Hatami, E.; Boya, V.K.N.; Kashyap, V.K.; Khan, S.; Hafeez, B.B.; Chauhan, S.C.; Jaggi, M.; Yallapu, M.M. miRNA-205 Nanoformulation Sensitizes Prostate Cancer Cells to Chemotherapy.Cancers 2018, 10, 289.
  • Yallapu, M.M.; Othman, S.F.; Curtis, E.T.; Gupta, B.K.; Jaggi, M.; Chauhan, S.C. Multi-functional magnetic nanoparticles for magnetic resonance imaging and cancer therapy. Biomaterials 2011, 32, 1890–1905.
  • Hao, L.; Patel, P.C.; Alhasan, A.H.; Giljohann, D.A.; Mirkin, C.A. Nucleic acid-gold nanoparticle conjugates as mimics of microRNA. Small 2011, 7, 3158–3162.
  • Neeraj Chauhan , Anupam Dhasmana Meena Jaggi Subhash C Chauhan , Murali M Yallapu miR-205: A Potential Biomedicine for Cancer Therapy. Cells. 2020; 9:1957. doi: 10.3390/cells9091957.

Round 2

Reviewer 1 Report

The manuscript includes more references after revision and explanation has been given to the sentences which are confusing. 

Reviewer 3 Report

Authors corrected paper according my suggestions.